

# A case study of a ducted gravity wave event over northern Germany using simultaneous airglow imaging and wind-field observations

Sumanta Sarkhel[1], Gunter Stober[2,3], Jorge L. Chau[2], Steven M. Smith[4], Christoph Jacobi[5], Subarna Mondal[1], Martin G. Mlynczak[6], and James M. Russell III[7]

[1]Department of Physics, Indian Institute of Technology Roorkee, Roorkee – 247667, Uttarakhand, India

[2]Leibniz-Institute of Atmospheric Physics, Schlossstr. 6, 18225 Kühlungsborn, Germany

[3]Institute of Applied Physics & Oeschger Centre for Climate Change Research, University of Bern, Sidlerstrasse 5, 3012 Bern, Switzerland

[4]Center for Space Physics, Boston University, Boston, Massachusetts, USA

[5]Institute for Meteorology, Leipzig University, Leipzig, Germany

[6]Atmospheric Sciences Division, NASA Langley Research Center, Mail Stop 420, Hampton, VA, USA

[7]Center for Atmospheric Sciences, Hampton University, 23 Tyler Street, Hampton, VA, USA

*Correspondence to*: Sumanta Sarkhel (sarkhel@ph.iitr.ac.in)





**Abstract**

An intriguing and rare gravity wave event was recorded on the night of 25 April 2017 using a multi-wavelength all‑sky airglow imager over northern Germany. The airglow imaging observations at

25    multiple altitudes in the mesosphere and lower thermosphere region reveal that a prominent upward propagating wave structure appeared in $O(^1S)$ and $O_2$ airglow images. However, the same wave structure was observed to be very faint in OH airglow images, despite OH being usually one of the brightest airglow emissions. In order to investigate this rare phenomenon, the altitude profile of the vertical wavenumber was derived based on collocated meteor radar wind-field and SABER temperature

30    profiles close to the event location. The results indicate the presence of a thermal duct layer in the altitude range of 85-91 km in the south-west region of Kühlungsborn, Germany. Utilizing these instrumental datasets, we present an evidence to show how a leaky duct layer partially inhibited the wave progression in the OH airglow emission layer. The coincidental appearance of this duct layer caused the wave amplitudes to diminish, resulting to exhibit as the faint wave front in the OH airglow

35    images during the course of the night over northern Germany.

**Keywords:** Airglow Imager, MMARIA meteor radar, Gravity Waves, Mesospheric Temperature Inversion, Thermal Ducting



## 1. Introduction

Multi-wavelength nighttime all-sky airglow imaging has become a widely used technique to retrieve valuable information of atmospheric gravity waves (GWs) as well as the dynamics of the mesosphere and lower thermosphere (MLT) region. GWs play a key role in the upper atmospheric dynamics because of their inherent properties of transferring momentum and energy from lower atmospheric regions to the middle and upper atmosphere (Fritts and Alexander, 2003). However, the ducting inhibits the vertical propagation of GWs and confines the major flow of wave energy and momentum to a rather limited altitude region (Chimonas and Hines, 1986). The earlier imaging studies using airglow emissions originating in the MLT region revealed different types of dynamical events like quasi-monochromatic GWs, ripples, mesospheric fronts (Taylor et al. 1995; Walterscheid et al., 1999; Hecht et al., 2001; Smith et al., 2003; Makhlouf et al., 1995; Bageston et al., 2011; Lakshmi Narayanan et al., 2012; Sarkhel et al., 2012; 2015a; 2015b; 2019; Hozumi et al., 2019; Mondal et al., 2021).

The characteristics and morphology of gravity wave events have been investigated for decades. Depending upon the characteristics and background conditions, the evolution of GWs in the atmosphere can be rather different. Large-scale waves can easily reach to the MLT region depending on their phase velocity compared to the background mean flow whereas small-scale waves are more susceptible to thermal and Doppler ducting (Walterscheid et al., 1999). The evolution of GWs and their interaction with the mean flow have been extensively studied using the linear theory of GWs. The waves can exert a significant amount of drag in the mean flow of the atmosphere and thereby play an important role in the middle atmospheric circulation. Instabilities present in the atmosphere do not support free propagation of GWs. The waves can break due to neutral instabilities into small-scale waves in the MLT region and generate secondary waves and ripple type structures due to body forces (Vadas et al., 2018; Becker & Vadas, 2018; Heale et al., 2020). Large-scale GWs can interact with the mean flow and generate one of the most puzzling mesospheric phenomena known as the mesospheric fronts (Dewan and Picard, 1998, 2001; Smith et al., 2005 and references therein). Mesospheric fronts can also propagate over long horizontal distances and therefore act as an efficient mechanism for transferring energy and momentum over long ranges with negligible energy loss in the atmosphere (Medeiros et al., 2018). Therefore, GWs propagation through a region of thermal/Doppler ducting can explain some of the properties of mesospheric fronts like the long-distance horizontal propagation.

The inhomogeneities in the background medium or the spatial (vertical and horizontal) structure of the temperatures and wind fields cause gradients in background temperature (static stability) and in the wind-field (dynamic stability), which confines the wave propagation. In particular, vertical gradients in temperature and wind field give rise to numerous interesting phenomena like wave reflection, wave ducts or waveguides in the MLT region. GWs can be ducted in a region where the vertical wavenumber ($m$) of the GWs is real ($m^2 > 0$) and the region is situated between two atmospheric altitude regions of imaginary vertical wavenumber ($m^2 < 0$). Once the GW falls into this ducted region, it gets trapped



because of repeated reflection from the bottom and upper layer (reflectance layers). However, the wave can freely propagate in a horizontal direction. If there is a reflection layer at a certain height, GWs can get reflected from this layer. The ducted wave is called thermally ducted or Doppler ducted (or both)

according to whether $m^2$ arises predominantly from the temperature gradient or the vertical shear of the horizontal wind (Walterscheid et al., 1999). A thermal duct is isotropic and will support ducted wave activity with any orientation, whereas a Doppler duct is very sensitive to the wave orientation (Fritts and Yuan, 1989). In this paper, we present a case study of a mesospheric wave structure using a multi-wavelength all-sky airglow imager and simultaneous measurements of 2D horizontally resolved wind-

field in the MLT region over northern Germany. Here, we present a first case study with the MMARIA meteor radar network in Germany (Stober an Chau, 2015; Stober et al., 2018) and co-located all-sky airglow imaging observations combining the available horizontally resolved wind and airglow information to infer the intrinsic GW parameters.

**2. Experimental Techniques**

A multi-wavelength all-sky airglow imager was procured from Boston University, USA and installed at Leibniz Institute for Atmospheric Physics, Kühlungsborn (54.11°N, 11.77°E), Germany. The imager has been operating since November 2016. The imager design is similar to an all-sky imager that is being operated at Padua Observatory, Asiago (Smith et al., 2017). The details of the imager and a few results

are available in Vargas et al. (2021). The imager features an Andor back-illuminated bare-CCD camera with $512 \times 512$ pixels resolution and a 16 mm fish-eye lens allows a maximum field of view of 180°.

The system is equipped with a temperature-controlled filter wheel that can record OH broadband emission (695–1050 nm) with a notch at 866.0 nm, Na emission (589.3 nm), $O_2$ emission (866.0 nm) and $O(^1S)$ emission (557.7 nm) in the MLT region. Based on rocket measurements, it has

been reported that these airglow emissions originate from layers of 8–11 km full width at half maxima (FWHM) or thickness with centroid height of around 86, 91, 94 and 97 km, respectively (Watanabe et al., 1981; Ogawa et al., 1987; Baker and Stair, 1988; Gobbi et al., 1992; Mende et al., 1993; Hedin et al., 2009). In addition, the imager also records thermospheric emission $O(^1D)$ (630.0 nm) from ~250 km altitude with around 40 km layer thickness (Sobral et al., 1992). The imager is also equipped with

a background filter in which the nightglow is minimal. This filter has a central wavelength of 605.0 nm and is used for photometric calibration of the images. In our investigation, we have used only emission originating from the MLT region.

The second data set used in this study is based on a meteor radar network operated in northern Germany known as MMARIA (Multi-static, Multi-frequency Agile Radar for Investigations of the

Atmosphere) (Stober and Chau, 2015). In this study, we used the Juliusruh meteor radar (54.6° N, 13.3°E) (e.g. Hoffmann et al., 2010) with a transmit power of 30 kW operated at 32.55 MHz, the Collm meteor radar (51.1°; 13.0°E) (e.g. Jacobi et al., 2007) with a transmit power of 15 kW operated at 36.2



MHz and a three multi-static passive systems installed at Kühlungsborn and Juliusruh. The bistatic
meteor detections from transmissions originating in Juliusruh and Collm are received
interferometrically at 32.55 MHz and 36.2 MHz, respectively. The horizontally resolved 2D wind-field
is based on the packed retrieval algorithm presented in Stober et al. (2018). Further, we restricted the
domain size to be slightly larger than the field of view of the airglow imager. The temporal resolution
of the 2D wind-field was 1 hour with a vertical resolution of 2 km at altitudes between 80 and 100 km.
The spatial grid for horizontally resolved winds is chosen to be 30 km × 30 km parallel to the Earth's
surface. All coordinates and radial velocities are corrected for projection errors using the WGS84 model
(National Imagery and Mapping Agency, 2000). An initial validation of the 2D wind-field retrievals,
and more details of the technique, can be found in Stober et al. (2018, 2021).

Another data set is the altitude profile of temperature that has been obtained from the SABER
instrument onboard TIMED satellite (Data source: http://saber.gats-inc.com; v2.0; Level 2A). The
retrieval of the ambient temperature at a given altitude is carried out using 15 µm emission from $CO_2$
molecules in the atmosphere. The location of the SABER measurement is less than 150 km from the
south-west corner of the imager FOV from where the wave entered. The uncertainty in the SABER
temperature retrievals is around ±3 K at 80 km, ±8 K at 90 km, ±1–2 K below 95 km and ±4 K at 100
km in the midlatitudes (García-Comas et al., 2008).

Figure 1 reveals the map of northern Europe where it shows the location of the multi-
wavelength airglow imager at Kühlungsborn, the Collm and Juliusruh meteor radar and the receiver
stations at Juliusruh and Kühlungsborn. The yellow box is the maximum horizontal coverage of the
airglow imager in the MLT region. The red asterisks are the SABER temperature measurement
locations. It is to be noted here that SABER 1 measurement location is quite far from the imager field
of coverage whereas, the measurement location of SABER 2 is close to the edge of the horizontal
coverage of the airglow imager.

## 3. Data Analyses

In order to retrieve scientific information, the raw images need to be processed. The standard method
of image processing including geospatial calibration, star removal and unwarping are available in the
literature (Garcia et al., 1997; Mondal et al., 2019 and references therein). In most situations, the
unwarped images are noisy and not very clear. It can be verified from the Figures 2-5 (a-f) that the
unwarped images are noisy for all the airglow filters. For the derivation of the wavelength, apparent
periodicity and phase velocity of the perturbation from the intensity fluctuation, the unfiltered unwarped
images may not be suitable. Therefore, in order to enhance the intensity perturbation by suppressing the
noise, the 2D FFT filtering techniques have been adopted from Mondal et al. (2019). In this filtering
technique, *Savitzky–Golay* and Gaussian window sizes are crucial. These window sizes are optimized
for each airglow emission filter which is discussed in Mondal et al. (2019) in detail. It may be noted



that the wave-like features in the filtered unwarped images are enhanced (shown in subplots of g-l of
Figures 2-5). These filtered unwarped images are utilized to derive the apparent phase velocity and
horizontal wavelength of the wave structure. In order to proceed further, a line is overlaid along the
direction of propagation for the particular number of consecutive images in which the wave structure
has been detected visually. This arrow (shown in subplots of g-l of Figures 2-5) represents the wave
vector of the structure. From the intensity variation along the line of propagation, the apparent phase
velocity and horizontal wavelength of the observed structure have been derived. The detailed
methodology is available in Mondal et al. (2019).

In order to investigate the vertical propagation characteristics of the observed wave structure
through the OH, Na, $O_2$ and $O(^1S)$ airglow emission layers, the altitude variation of squared vertical
wavenumber ($m^2$) needs to be computed. Following Nappo (2002), the relation for the vertical
wavenumber can be expressed as:

$$m^2(z) = \frac{N^2}{\hat{c}^2} - \frac{U_k{}'}{H\,\hat{c}} - k_x{}^2 - \frac{1}{4H^2}. \qquad (1)$$

Here, $k_x$ is the horizontal wavenumber, $H$ is the scale height, $U_k$ is the projected wind along the wave
vector, $U_k{}'$ is the vertical wind shear, $c$ is the observed phase velocity and $\hat{c}$ is the intrinsic phase velocity
($c - U_k$). The altitude profile of the projected wind along the wave vector has been calculated from
the altitude variation of the 2D horizontally resolved wind field within the observed structure. In
addition, the 2D wind field at the centroid height of each airglow emission are overlaid on the filtered
unwarped airglow images (shown in subplots of g-l of Figures 2-5). This approach gives us the
opportunity to calculate the horizontal wind velocity within the observed wave structure more precisely.

**4. Results**

Figures 2-5 depict all-sky airglow imaging observations at four different airglow emissions originating
at the MLT region along with the MMARIA 2D horizontally resolved wind-field measurements over
northern Germany during the cloudless and moonless night of 25 April 2017. Figures 2(a-f) show the
sequence of unwarped images observed in $O(^1S)$ airglow emission. Figures 2(g-l) depict the
corresponding 2D FFT filtered images overlaid on the MMARIA horizontal 1 hour averaged wind field
in two dimensions at the centroid height of $O(^1S)$ airglow emission in the MLT region. In a similar
manner, the upper horizontal subplots in Figures 2-4 represent the sequence of unwarped images
observed in $O_2$, Na and OH band emissions, respectively. The bottom subplots represent their
corresponding 2D FFT filtered images overlaid on the MMARIA 2D horizontally resolved wind field
at the centroid height of the respective airglow emissions. Figures 2-5 depict the existence of a front-
wall with small undulation wave structure. The horizontal propagation of the observed wave structure
is from south-west to north-east and entered within the field-of-view (FOV) of the imager over northern
Germany. The wave structure entered at the south-western edge of the FOV of the imager at around



21:35 UT. Although we have shown the sequence of images (Figures 2-5) from 21:56 UT onward when
the wave structure is fairly visible in the south-west corner of the imager FOV. The wave structure is
very prominent in both $O(^1S)$ and $O_2$ images. On the other hand, it is very faint in Na and OH images.
It is interesting to note that the measurement location of SABER 2 falls in the path of the propagation
of the wave structure.

In order to investigate the vertical phase propagation of the wave structure, the intensity of
slices of $O(^1S)$, $O_2$ and Na images were plotted at a given time in the direction of propagation and they
have been horizontally shifted based on its deduced phase speed of the leading front, to account for the
acquisition time differences between each image. The mean phase velocities deduced from $O(^1S)$, $O_2$
and Na images are 50.2 ± 4.8 m/s, 46.6 ± 4.2 m/s and 44.5 ± 9.7 m/s respectively.  Since the wave
structure appeared to be very faint in the OH images, observing the GWs in these images becomes
practically impossible. A key element in the analyses is the horizontal vs. altitude intensity plot to
determine the vertical phase propagation of the observed GW similar to Smith et al. (2005). The green,
red and orange lines in each subplot of the Figures 6 (a-d) depict the intensity variation of $O(^1S)$, $O_2$
and Na airglow emissions respectively along their line of propagation vector considering the large-scale
vertical wind shear and, thus obtaining the intrinsic phase progression, which is actually needed to
distinguish the vertical propagation direction. The slices have been separated vertically in order to
represent the assumed relative vertical separations of the emission layers in the MLT region corrected
for the observed phase speed of the wave. We can clearly observe that the intrinsic phase in the $O_2$
airglow (centroid emission height: 94 km) is always lagging $O(^1S)$ airglow (centroid emission height:
97 km). Thus, the lagging phase front in $O_2$ airglow, which is in the lower height, as compared to the
$O(^1S)$ airglow indicates that the phase front appeared in $O(^1S)$ and then $O_2$ airglow emission layer.
Therefore, the vertical phase progression of the wave structure is downward. Hence, based on this
information, we can conclude that we have captured an event of an upward propagating wave in $O(^1S)$
and $O_2$ airglow emission layers on the night of 25 April 2017 over northern Germany. However, the
intrinsic phase in the Na airglow (centroid emission height: 91 km) appears to lead the $O_2$ airglow,
indicating that the vertical phase progression of the wave structure is different in the Na airglow
emission layer as compared to $O(^1S)$ and $O_2$ airglow emission layers.

In order to find the vertical propagation characteristics of the wave structure, the derivation of
the altitude profile of $m^2$ is required with the knowledge of the vertical temperature profiles and the
projected wind along the wave vector. In addition, it is clear from Equation (1) that we also need to
calculate the intrinsic phase velocity of the wave structure. Hence, the horizontally spatially-averaged
projected wind profile within the region of wave structure and along the wave vector have been
calculated and the hourly values for 21:30-22:30 UT are shown in the Figure 7a. Figures 7b and 7c
depict the altitude profiles of SABER temperature along with the measurement uncertainty plotted as
horizontal error bar over the SABER 1 and 2 locations (refer to Figure 1) respectively. It is to be noted
that the SABER temperature measurements were carried out around 1 hour prior to the event and we





didn't observe any wave activities during this period. Hence, there will not be any significant differences in the temperature and this temperature profile has been used to calculate the altitude profile of $m^2$. Following Equation (1) and based on the SABER temperature profile, imager, and MMARIA projected wind profile measurements, we have calculated the altitude profile of $m^2$. Figures 7d and 7e are the

altitude profiles of $m^2$ along with the proportional error plotted as horizontal error bar over SABER 1 and 2 locations respectively. Both $m^2$ profiles have been derived using the same projected wind shown in Figure 7a. It is interesting to note that the temperature profile doesn't show any inversion at the SABER 1 location. However, we can observe the mesospheric inversion layer (MIL) at the SABER 2 location. The $m^2$ profile at SABER 2 reveals that a duct region exists in the altitude range of 85-91 km

where positive $m^2$ is vertically sandwiched between the negative $m^2$ values. In order to find the nature of the duct layer, we have analyzed the contribution of each term of the dispersion relation (Equation 1). The contribution terms in $m^2$ profiles are like ~89.5 % from the buoyancy term (thermal gradient) $\left(\frac{N^2}{\hat{c}^2}\right)$, ~6.5 % from the wind shear term $\left(\frac{U'_k}{H\hat{c}}\right)$, ~1.9 % from squared of horizontal wavenumber $k^2$ and ~2.1 % from scale height term $\left(\frac{1}{4H^2}\right)$. The contribution suggests that the buoyancy term (thermal

gradient) dominates in the $m^2$ profile whereas wind plays an insignificant role in the $m^2$ profile. The wind profile does not show any drastic change within the duct layer (Figure 7a). Therefore, the duct layer observed in the MIL region at 85-91 km is a *thermal duct* by nature, dominated by the thermal gradient of the MIL which is observed merely 1 hour before the appearance of the wave fronts in the south-west region of imager FOV. It is worth mentioning that the observed MIL is close to the edge of

the horizontal coverage of the airglow imager and coincide with the path of the propagation of the wave structure (refer to Figures 2-5).

## 5. Discussion

As discussed above, the multi-wavelength all-sky airglow imager installed at Kühlungsborn, Germany

can record images at four airglow emissions originating from the MLT region of the Earth's atmosphere. Our main objective is to combine data from the all-sky imager to investigate the spatial information of the waves in the MLT region and corroborate with the 2D horizontally resolved wind field at different altitudes using MMARIA. The optical and radio measurements in 2D gives us a unique opportunity to investigate intriguing wave events in the MLT region over northern Germany. We have captured a

front-like wave structure during the all-sky airglow imaging observation on 25 April 2017 at O($^1$S), O$_2$, Na and OH airglow emission layers. As discussed in the Results section, we captured an upward propagating wave structure and it appeared to be very prominent in both O($^1$S) and O$_2$ images propagating from south-west to north-east. On the other hand, it is faintly observed in both Na and OH images. As the Na airglow is a relatively weaker emission compared to O($^1$S) and O$_2$ airglow, it may

be the plausible reason for observing faint structure in Na airglow images. Hence, it is expected that any wave structure observed in that filter may not be prominent due to poor signal-to-noise ratio.





However, it is well-known that the OH airglow emission is much brighter than $O_2$ and $O(^1S)$ emissions in the MLT region and has been widely used for the investigating of GWs (e.g. Taylor et al., 1995; Yamada et al., 2001; Mukherjee, 2003; Li et al., 2005; Yue et al., 2010). Hence, any wave structure
should be very prominent in the OH airglow images. In fact, clear signatures of wave activities were observed on other nights in OH airglow images over northern Germany. However, the wave structure, on 25 April 2017, can barely be observed in the OH airglow images. The detailed analyses of the plausible cause behind the occurrence of this rare event have been carried out and discussed below:

*5.1 Reduction in the OH airglow emission intensity*

As it is mentioned above that the OH airglow emission is normally a strong emission for the investigation of dynamics in the MLT region. However, reduction in the overall density of H atoms and $O_3$ molecules, that are reacting to form the excited OH molecule, could have led to the weak OH airglow emission on that night. Consequently, the wave structure observed in the OH filter may not appear as
prominent as it should be, due to poor signal-to-noise ratio. In order to explore the possibilities, we have carried out comparison of the SABER measured altitude profiles of H atomic, $O_3$ molecular densities and OH Volume Emission Rate (VER) for the previous & following nights close to the SABER measurement location (figures not shown). The SABER measurements indicated that the centroid height of OH airglow emission occurred near 87 km. We found that there are no significant differences
of the density of H & $O_3$ and OH VER profiles. Hence, the fact that we observed the faint wave structure in the OH airglow image is not due to the reduction in the overall intensity level of OH airglow emission on that night. We believe that the perturbation wave amplitude will finally decide whether the structure observed in the OH airglow image is faint or not.

*5.2 Cancellation of waves in the OH airglow emission layer*

Based on model calculation, Liu and Swenson (2003) reported that for vertically propagating GWs, the amplitude of airglow perturbations observed using ground-based measurements is larger for longer vertical wavelength, due to the smaller cancellation effect within each layer. This cancellation factor was introduced by Swenson and Gardner (1998) for OH airglow to relate the observed vertically-
integrated airglow intensity perturbations to the wave amplitudes. This cancellation factor causes a GW that may not be observable in an airglow emission layer, when observed using ground-based airglow instruments, if the vertical wavelength is less than the FWHM. (e.g., Swenson and Gardner, 1998, Liu and Swenson, 2003, Vargas et al, 2007).

Based on the sounding rocket measurements, it has been reported that the OH, Na, $O_2$ and $O(^1S)$
airglow emission layers are originating in the MLT region with FWHM of 8-10 km (Watanabe et al., 1981; Ogawa et al., 1987; Baker and Stair, 1988; Gobbi et al., 1992; Hedin et al., 2009). On many occasions, these in-situ rocket borne measurements reveal that the FWHM of the $O_2$ emission layer is





more than the one of OH whereas, the FWHM of Na airglow layer is nearly comparable with that of OH and $O_2$. However, in general, the FWHM of $O(^1S)$ airglow layer is 1 or 2 km less compared to the

OH and $O_2$ airglow layers. In the present case, the vertical wavelength of the wave structure is in the range of 8-11 km and hence the cancellation factor has a similar effect in OH and $O_2$ airglow layers. Therefore, it is unlikely that the faint wave structure observed in the OH airglow images is due to the cancellation of waves in the OH airglow emission layer. It appears that this rare event occurred because of conducive background condition over northern Germany on that night.


### 5.3 Thermal ducting of waves

In order to investigate this interesting and intriguing observation, the derivation of the altitude profiles of $m^2$ has been carried out over both SABER 1 and 2 measurement locations. Figure 7 suggests the presence of a thermal duct layer (85-91 km) merely 1 hour before the appearance of the wave fronts in

the south-west region of imager FOV. In order to demonstrate the event, we have drawn a schematic diagram depicted in Figure 8 (not to scale) from the results of the Figures 6 and 7. It is well-known that the MIL tends to be quite stable within a few hours' time scale (Meriwether and Gerrard, 2004). However, they can be spatially discontinuous as observed from SABER 1 and 2 temperature profiles. The theory of GW propagation predicts that the waves should reflect if the vertical wavenumber has an

imaginary component. Hence, if $m^2 < 0$, then the transmitted wave will die out because of its amplitude decreases exponentially. It creates a region of evanescence in which the wave cannot vertically propagate and is reflected (Isler et al., 1997; Fritts and Yuan, 1989; Hines and Tarasick, 1994; Huang et al., 2010). Figure 8 shows how the wave structure freely propagated in three dimensions and it encountered the bottom of the duct layer at 85 km. The $m^2$ indicates that the existence of a weak

evanescent region at 85 km. Therefore, the duct layer observed in the present case is a "leaky duct" from the bottom side. Thus, some of energy of the wave could penetrate through the bottom of the duct layer. Thus, a few wave fronts travelling from south-west to north-east could partially enter from the bottom of the duct layer in the imager FOV in Na and OH airglow emission regime. The other part of the wave structure, which did not encounter the duct layer, freely propagated and entered in the FOV

of the imager at $O(^1S)$ and $O_2$ airglow emission layers (demonstrated in the Figure 8) situated at higher altitudes. This is supported by the phase progression analyses shown in Figure 6, wherein we captured the wave structure as 'upward propagating wave' in $O(^1S)$ and $O_2$ airglow emission layers on that night. However, the intrinsic phase of the wave structure in the Na airglow emission layer indicates that the vertical phase progression is different as compared to the $O(^1S)$ and $O_2$ airglow emission layers. This

difference in the vertical phase progression is caused by the combined effect of wave structure within and beyond the ducted layer as the airglow imager captured vertically integrated Na airglow emission intensity peaked at 91 km with a FWHM of ~10 km (Mende et al., 1993). On the other hand, the intensity in the OH airglow images appeared so faint that the determination of the phase progression of the waves in these images becomes practically impossible. The multi-instrument observation gave us the



opportunity to investigate the plausible physical process behind this intriguing and rare phenomenon where the thermally leaky ducted layer partially inhibited the wave progression in the OH airglow emission layer. The coincidental appearance of the duct layer caused the wave amplitudes to diminish, resulting the OH airglow images to exhibit as the faint wave front on the night of 25 April 2017 over northern Germany.

## 6. Summary and Conclusion

A case study of a wave event observed on the night of 25 April 2017 over northern Germany using optical and radar instruments is reported here. The multi-wavelength all-sky airglow imager recorded an upward propagating wave structure at multiple airglow altitudes in the MLT region. It appeared to be very prominent in $O(^1S)$ and $O_2$ airglow images. However, the same wave structure is observed to be faint in both Na and OH airglow images, despite OH being one of the strong airglow emissions. In order to investigate this intriguing phenomenon, the derivation of the altitude profiles of $m^2$ was carried out using collocated MMARIA altitude profile of the horizontally resolved wind field and a SABER temperature profile close to the event location. The obtained $m^2$ profiles indicates the presence of a thermal duct layer in the altitude range of 85-91 km in the south-west region of Kühlungsborn. The leaky thermal duct layer allowed the wave structure travelling from south-west to north-east that could partially enter from the bottom of the duct region (at 85 km) in the imager FOV in the OH airglow emission layer. Whereas, the other part of the wave structure, which did not encounter the duct layer, freely propagated and entered in the FOV of the imager at $O(^1S)$ and $O_2$ airglow emission layers. The appearance of this duct layer caused the wave amplitudes to diminish, resulting the weak wave structure observed in the OH airglow images on that night over northern Germany.

## 7. Code/Data availability

The SABER temperature data is available in http://saber.gats-inc.com. The airglow imager and MMARIA wind radar data are available at https://doi.org/10.5281/zenodo.4925828.

## 8. Author contribution

SS conceived the theme, devised the data processing methods, carried out data analyses, and wrote the manuscript. GS, JLC, SMS helped in conceiving the theme and contributed in writing the manuscript. SM helped in carrying out the data analyses and contributed in writing the manuscript. CJ, MGM, JMR contributed in writing the manuscript.

## 9. Competing interests

Competing interests: Gunter Stober is one of the topical editors and Christoph Jacobi is one of the Editors-in-Chief. The authors declare that they have no conflict of interest.





**Acknowledgements**

S. Sarkhel acknowledges the Leibniz Institute of Atmospheric Physics for hosting him during the summer of 2017. S. Mondal acknowledges the fellowship from the Ministry of Education, Government of India for carrying out this research work. S. Sarkhel thanks Amitava Guharay, Fazlul I. Laskar, and
Jean-Pierre St-Maurice for useful discussion. The help from Govind Gaur in calculating the uncertainty in $m^2$ is duly acknowledged. This work is partially supported by the WATILA project (SAW-2-15-IAP-5 383) funded by Leibniz-Gemeinschaft. Funding for S.M. Smith was provided by NSF award AGS-1909237. G. Stober receives support from University Bern, the Oeschger Center for Climate Change Research (OCCR) and by the ARISE2/ARISE-IA project (www.ARISE-project.eu) from the European
Community's Horizon 2020 programme. This work is also supported by Ministry of Education, Government of India.



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

**Figures**

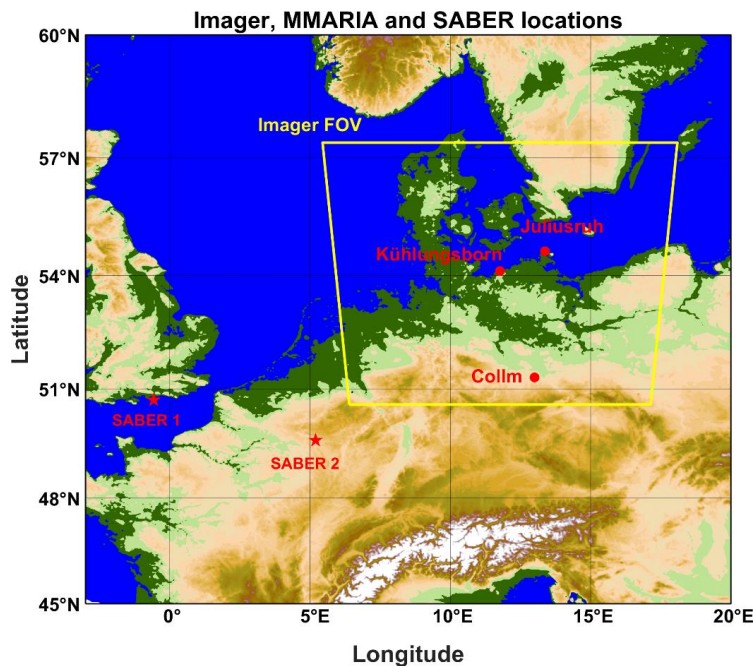

**Figure 1:** The map of northern Europe where it shows the location of the multi-wavelength airglow imager at Kühlungsborn, MMARIA transmitter stations (Collm and Juliusruh) and receiver stations (Juliusruh and Kühlungsborn). The map has been generated from ETOPO1 1 Arc-Minute Global Relief Model (Amante and Eakins, 2009). The yellow box is the maximum horizontal coverage of the airglow

imager in the MLT region. The red asterisks show the SABER temperature measurement locations.

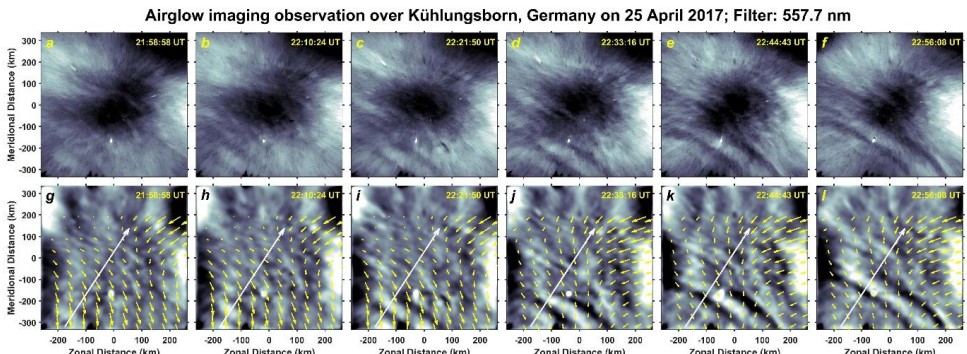

**Figure 2:** (a-f) Sequence of O($^1$S) 557.7 nm airglow images on 25 April 2017 over northern Germany. (g-l) Corresponding 2D FFT filtered images overlaid on horizontal variation of wind-field (yellow arrows) at the emission centroid height (97 km) measured by MMARIA. The white arrow denotes the wave vector.

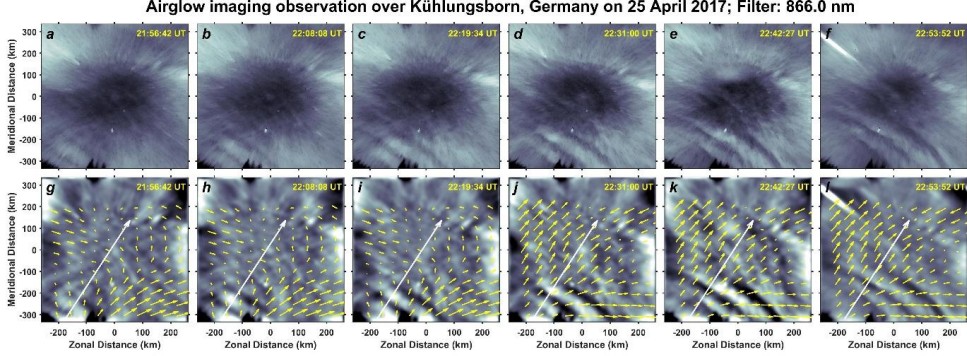

**Figure 3:** (a-f) Sequence of O$_2$ 866.0 nm airglow images on 25 April 2017 over northern Germany. (g-l) Corresponding 2D FFT filtered images overlaid on horizontal variation of wind-field (yellow arrows) at the emission centroid height (94 km) measured by MMARIA. The white arrow denotes the wave vector.

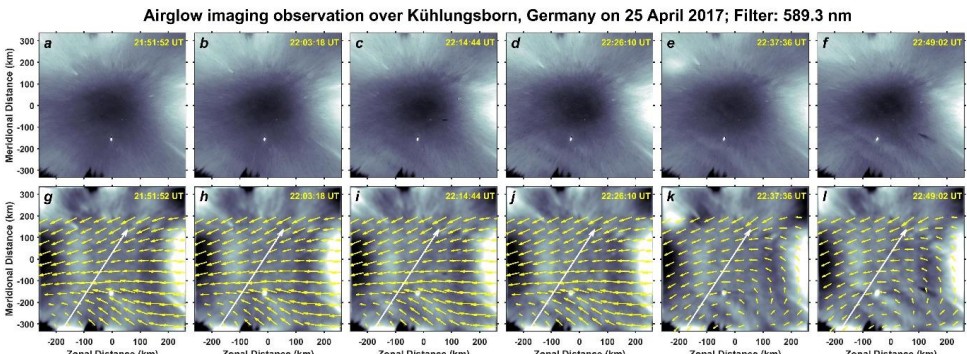

**Figure 4:** (a-f) Sequence of Na 589.3 nm airglow images on 25 April 2017 over northern Germany. (g-l) Corresponding 2D FFT filtered images overlaid on horizontal variation of wind-field (yellow arrows) at the emission centroid height (91 km) measured by MMARIA. The white arrow denotes the wave vector.

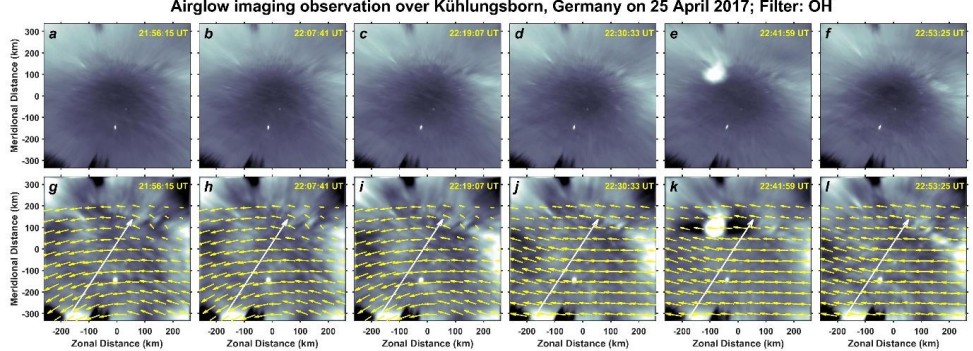

**Figure 5:** (a-f) Sequence of OH airglow images on 25 April 2017 over northern Germany. (g-l) Corresponding 2D FFT filtered images overlaid on horizontal variation of wind-field (yellow arrows) at the emission centroid height (86 km) measured by MMARIA. The white arrow denotes the wave vector.



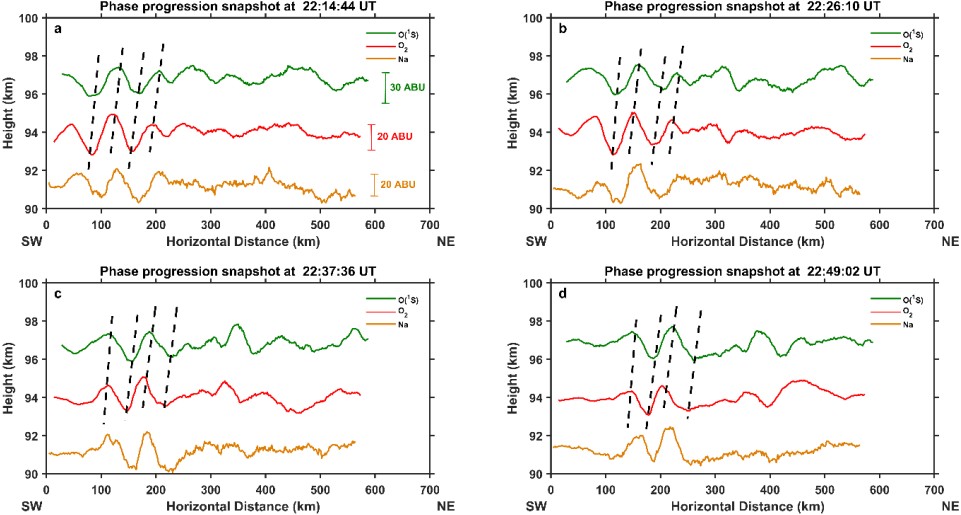

**Figure 6:** (a-d) The green, red and orange solid lines in each subplot depict the wind shear corrected intensity variation of O($^1$S), O$_2$ and Na airglow emissions respectively along their line of propagation. The slices have also been horizontally shifted using the intrinsic horizontal wave speed in order to make a quasi-simultaneous ''snapshot'' of the vertical structure of the wave field. The slices have been separated vertically in order to represent the relative vertical separations of the emission layers in the MLT region. The emission brightness is given in arbitrary brightness units (ABU) in panel (a) and refers to all four panels.



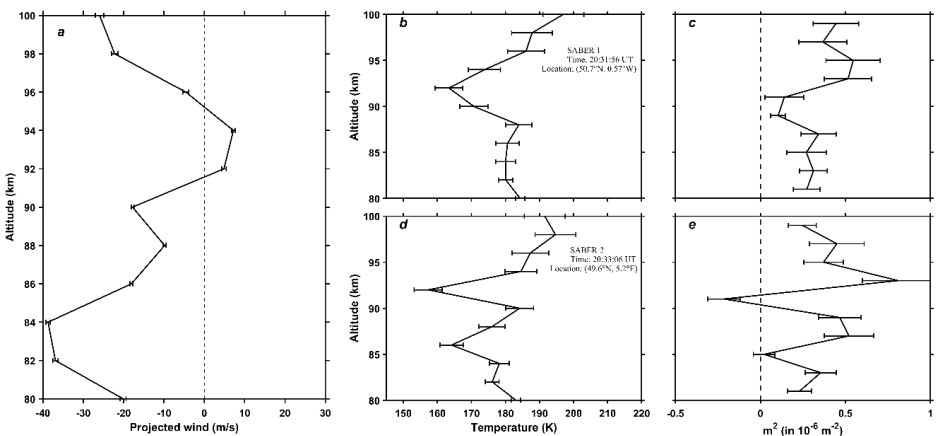

**Figure 7:** (a) The altitude profile of projected horizontal wind along the wave vector measured during
21:30 - 22:30 UT. (b-c) The altitude profiles of SABER temperature along with measurement
uncertainty plotted as horizontal error bar over SABER 1&2 locations respectively. (d-e) The altitude
profiles of $m^2$ derived using the Equation (1) along with the proportional error plotted as horizontal
error bar over SABER 1&2 locations respectively.





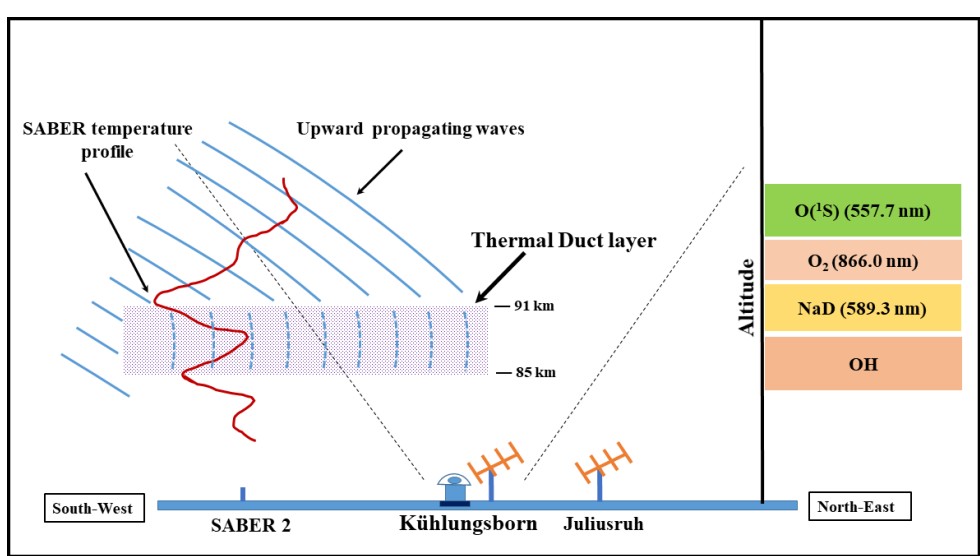

625

**Figure 8:** Schematic diagram of ducted wave fronts in OH and Na airglow emission layers over northern Germany on 25 April 2017.