# Peer review of "A case study of a ducted gravity wave event over northern Germany using simultaneous airglow imaging and wind-field observations"

_Annales Geophysicae, 2021_

## Author Response (AR1)

**Reply to the Reviewer 1**

**Manuscript #: angeo-2021-48**

**Title: A case study of a ducted gravity wave event over northern Germany using simultaneous airglow imaging and wind-field observations**

**The work combines Airglow imager and Meteor radar observations from Germany and uses SABER measurements to bring out/discuss a special gravity wave event that leaves its footprint in oxygen green line, O2 emissions but not in OH emission. The authors propose the formation of a leaky duct to explain the observations. While this is a realistic possibility, other factors need to be ruled out to strengthen the argument. The observation of this wave activity in Na emission is also enigmatic and require more attention. The observations reported merit publication but the authors need to critically address a few issues described below.**

Reply: We thank the reviewer for the appreciation and critical comments which have improved the content of the manuscript significantly. The responses to the reviewer's comments are provided below in the same sequence.

**On dataset and techniques: Does this imager consist of 512 × 512 pixels ANDOR back-illuminated CCD camera or 1k × 1k camera as mentioned by Vargas et al. (2021)? For completeness, provide details on the bandwidth of the interference filters. Is there a possibility of contamination of OH broadband emission from other mesospheric lines? How do the authors rule that this contamination is not responsible for the lack of wave signatures in the OH band? What are the integration times used for these observations?**

Reply: Yes, it is the same 1k × 1k CCD camera mentioned in Vargas et al. (2021). On the night of 25 April 2017, the images were binned 2 × 2 (in real time) in order to achieve better signal-to-noise ratio. The bandwidth of the 589.3 nm, 866.0 nm, and 557.7 nm filters is 2.0 nm whereas, the OH filter is a broadband (695–1050 nm). Hence, there is no contamination of OH broadband emission from other mesospheric lines (557.7 and 589.3 nm).

The nearest lines of OH ($X^2\Pi$) are at P1(6) and R(7,3) (854.9 and 877.1 nm respectively) that are quite far-off from $O_2$ emission line (866.0 nm) (Chamberlain, 1961). In addition, the wave signatures in the OH (when detected) and $O_2$ images differed in both morphology and phase. Hence, we are confident that no significant broadband OH occurred within the 866.0 nm filter bandwidth. The integration times for the 589.3 nm, 866.0 nm, and 557.7 nm images were 120s and 15s for the OH filter. We have included this information in the revised manuscript. (Page No. 4-5; Line No. 92-93, 104-113)

**Figures 2-5: Why does the central dark spot appear in all the images? Detector issue? How does the 2D FFT filtering remove this feature?**

Reply: The central dark spot appearing in every all-sky airglow image is due to the van Rhijn effect; a combination of the finite width of the emission layer and the variation of the line of sight through the layer with increasing zenith distance. There was no issue with the CCD detector. We have mentioned this in the revised manuscript (Page No. 7; Line No. 185-188).

We have used 2D FFT filtering techniques adopted from Mondal et al. (2019). In this filtering technique, the Gaussian filter window behaves as a low-pass filter and therefore chosen to remove high frequency noises. In the spatial domain, the filtered image is the convolution of an unfiltered image and the Gaussian filter. However, in the power spectral domain, we can simply multiply the 2D FFT power spectra of the unfiltered image with the 2D power spectra of Gaussian filter window by positioning it at the peak-power position of the 2D FFT spectra of the image and then perform the inverse 2D FFT. The airglow image contains range of spatial scale sizes instead of monochromatic of gravity waves. Hence choosing the proper window size of Gaussian filter, we can select the desired scale sizes of gravity waves and the filtered images get enhanced with those features. In this process, the central dark spot is automatically removed.

**Science issues: Figures 2-5: The wind fields are widely different even within the "ducted" layer. Why? This is contrary to the argument (used later for Figure 7a) that the profile does not show any drastic change within the ducted layer.**

Reply: We wanted to convey that the thermal gradient dominates in the $m^2$ profile whereas the wind shear doesn't play any significant role. We have removed the sentence "The wind profile does not show any drastic change within the duct layer (Figure 7a)." in order to avoid any confusion (Page No. 9; Line No. 260-261).

**Figure 6: Why is the downward phase progression not seen in the Na airglow intensity? The authors have mentioned 91 km as the emission height of this emission and 85-91 km as the altitude extent of the thermal duct. The intensity variation at this wavelength also seems to be different at larger horizontal distance.**

Reply: The upper half of the Na airglow emission is situated above the ducted layer which is the free upward propagating region of waves. However, the bottom half of layer lies in the ducted region (as sketched in Figure 8) wherein the wave is confined to propagate horizontally and not in the upward direction. Therefore, the difference in the vertical phase progression in the Na airglow intensity is caused by the combined effect of wave structure within and beyond the ducted layer as the airglow imager captured vertically integrated Na airglow emission intensity. Whereas, the $O(^1S)$ and $O_2$ airglow emission layers are beyond the ducted layer and hence clearly shows the downward phase progression.

Yes, we agree with the reviewer that the intensity variation at this wavelength also seems to be different at larger horizontal distances. However, we are unable find out the cause based on the present dataset. We have mentioned this in the revised manuscript. (Page No. 8; Line No. 231-233).

**I feel that the sub-units of Figure 7 are wrongly described in the text. For example, 7c and 7e (instead of 7d and 7e) show variations in m^2. Same for Figures 7b and 7c.**

Reply: We thank the reviewer for pointing out these typos. We have corrected it in the revised manuscript (Page No. 8; Line No. 239, 249).

**Is $m^2$ negative at 85 km? How important is the negative $m^2$? If it is less but positive (at 85 km), how efficient is the ducting? This is an important issue as the authors propose a leaky duct here. If there is leakage each time the wave is reflected from the edges of the duct, how does the Na intensity variation show similar intensity variation but different phase progression (at least at shorter horizontal distances) in Figure 6? Will the original and reflected waves inside the duct not interfere? How is the spatial coherence maintained? This is also an important issue given the different wind fields at the airglow emission altitudes. The authors need to discuss these issues.**

Reply: The $m^2$ profile at SABER 2 reveals that a duct region exists in the altitude range of 85-91 km where positive $m^2$ is vertically sandwiched between a negative $m^2$ value (at 91 km) and a less positive value (at 85 km). The $m^2$ profile indicates that the existence of a weak evanescent region at 85 km (as $m^2$ is not negative). Therefore, the duct layer observed in the present case is a "leaky duct" from the bottom side. Hence, some of the energy of the wave could penetrate through the bottom of the duct layer and other part of the energy will be reflected back downward. The penetrated wave got trapped inside the duct layer and could only travel horizontally.

As we have already discussed above that the upper half of the Na airglow emission is situated above the ducted layer which is the free upward propagating region waves. However, the bottom half of layer is lies in the ducted region wherein the wave is trapped and don't any propagate in the upward direction. Therefore, the difference in the vertical phase progression in the Na airglow intensity is caused by the combined effect of wave structure within and beyond the ducted layer as the airglow imager captured vertically integrated Na airglow emission intensity.

The penetrated wave only entered the leaky duct layer from the bottom side. The reflected wave from the edge of the bottom of the duct layer propagated downward below and didn't enter the duct layer. Therefore, there won't be any interference between the original and the reflected waves inside the duct layer.

The winds are analyzed as an average along the propagation path of the wave fronts. The spatial wind retrievals imply a large spatial coherence of about 60 km (the correlation length is set to include the next grid cell) and a temporal correlation of about 30 minutes. Thus, a certain degree of coherence is an essential part of the wind analysis for this case. Furthermore, due to the thermal wind balance, changes in the vertical temperature profile led to corresponding changes of the winds at the different altitudes. The spatial variability of the wind field is also indicated by the bending of the wave fronts found in Figure 2 and 3, whereas the wind fields remain stable at the altitude of the OH emission line presented in Figure 5. However, due to the missing spatial information of temperature observations, it is not possible to understand how this spatial variability affects also the leakage of the duct. We have discussed this in the revised manuscript. (Page No. 7; Line No. 193-202).

**Reply to the Reviewer 2**

Manuscript #: angeo-2021-48

Title: A case study of a ducted gravity wave event over northern Germany using simultaneous airglow imaging and wind-field observations

**Overview**

**This paper is about multi-wavelength airglow observations of a gravity wave event above northern Germany. The peculiar finding is a strong wave signature visible in the $O(^1S)$ (97 km altitude) and $O_2$ (94 km altitude) bands, a fainter signature in the Na band (91 km altitude), and no wave signature in the OH band (86 km altitude). Auxiliary data used in the study are wind measurements by a meteor radar network and temperature profiles acquired by the SABER instrument on the TIMED satellite. To my knowledge, this is the first report on such a wave event with co-located local measurements of horizontal wind. Information on horizontal wind is of particular importance for the estimation of intrinsic wave parameters. Using the available measurements, the authors derive a vertical wavenumber profile from the gravity wave dispersion relation and conclude that a thermal duct is responsible for the non-detection of wave signatures in the OH band.**

**The paper is well written and presents some novel results. I recommend it for publication in Annales Geophysicae subject to the authors addressing the comments below. My main concern is the lack of a plausible explanation for the non-detection of wave signatures in the OH band (see major comment below). Without an explanation, the paper is merely a compilation of observations. Although the uniqueness of the presented observations may justify their publication (I leave this to the editor), a proper description, discussion and evaluation of potential mechanisms responsible for the diminishing modulation in OH airglow brightness will greatly increase the scientific value of this paper.**

Reply: We thank the reviewer for the appreciation and critical comments that have improved the content of the manuscript significantly. The responses to the reviewer's comments are provided below in the same sequence.

**Major comment**

**I am surprised that the authors did not use Fig. 6 to estimate the vertical wavelength. From the slopes of the dashed lines I get values ranging between 8 and 12 km. On the other hand, $m^2 = 0.5$ (taken from Fig. 7e) leads to lambda_z = 8.8 km. This value is consistent with the previous estimate and thus increases confidence in the derived vertical wavelengths. Also, from Fig. 7e I estimate the width of the duct h=5.5 km. According to lambda_z = 2h/n (see e.g., Dong et al. 2021; reference id given below) we can conclude that the wave propagating within the duct is likely of $0^{th}$ order (n=1) assuming that the vertical wavelength inside the duct is approximately the same as above the duct. Indeed, based on Fig. 7e, $m^2$ values at 88 km (center of the duct) and at 95 km (above the duct) are similar. However, the $0^{th}$ mode is a symmetric mode and, because the center of the duct (~88 km) is approximately aligned with the center of the OH layer (~86 km), the ducted wave should result in detectable**

**modulations in airglow brightness. Some pieces of the explanation of the non-detection of wave signatures in OH airglow are clearly missing here. On the other hand, the authors provide no explanation either. In my opinion the sentence "The coincidental appearance of the duct layer caused the wave amplitudes to diminish" (l.327) is unsupported speculation. Before making such a statement, the authors should at least discuss conditions which *can* result in cancellation of wave signatures in OH airglow due to the viewing geometry or otherwise.**

Reply: We thank the reviewer for the suggestion to estimate the vertical wavelength from Figure 6 and have included this aspect in the revised manuscript (Page No. 10-11; Line No. 322-329).

     The waves were visible in the OH airglow images during the course of the event, but were faint compared to $O(^1S)$ and $O_2$ emissions. Dong et al. 2021 discussed the various wave modes that can exist in a ducting region. The vertical wavelength of the ducted wave ($\lambda_z$) and the duct width ($h$) are related via the relationship of $\lambda_z = 2h/n$ ($n = 1, 2, 3, \ldots$ denotes 0th, 1st, 2nd wave mode). Using Figure 7e, we calculated the duct thickness to be 6 km and the vertical average wavelength in the OH emission layer to be 11.5 km. Hence from the above relation, we can conclude that the 0th wave mode ($n = 1$) can exist in the duct.

     We suggest that the ducting layer (85-91 km) observed in the present case (Figure 7c & e) was weak and so inhibited free propagation within that particular altitude region. Only part of the energy of the wave could penetrate through the bottom of the duct layer and the other part of the energy will be reflected back downward. Being localized, the part of the wave continued to propagate upwards into the $O_2$ and $O(^1S)$ emission layer at higher altitudes (represented schematically in Figure 8). Since only a part of the wave energy could enter in the duct region (85-91 km), the wave structure was observed to be faint in the OH airglow images.

     We agree that the sentence "The coincidental appearance of the duct layer caused the wave amplitudes to diminish" is somewhat confusing and we have modified it in the revised manuscript (Page No. 12; Line No. 367-368).

**Minor comments**

**l. 50: It would be helpful if you could define "large-scale waves"**

Reply: In this context, we meant "large-scale waves" as those with several tens of kilometers horizontal wavelength. It has been included in the revised manuscript (Page No. 3; Line No. 50).

**l. 55: "Instabilities present in the atmosphere do not support free propagation of GWs." This sentence is not clear to me. The assertion implies that instabilities are a property of the (background) atmosphere. However, the question whether an instability arises also very much depends on the properties of a particular wave propagating through the atmosphere. For instance, depending on the wind profile, an eastward propagating wave may propagate freely whereas a westward propagating wave is filtered due to a critical level.**

Reply: We agree with the reviewer that this sentence is somewhat confusing. We have removed it.

**l. 56: This may be a discussion about terminology, but it is my understanding that wave breaking is a process during which a wave breaks down and energy is transferred to smaller eddies. I wouldn't call these eddies waves because the word 'waves' implies some form of coherent structure. The generation of secondary waves is a separate process.**

Reply: We have modified the sentence in the revised manuscript (Page No. 3; Line No. 56-57).

**l. 65: Please explain "inhomogeneities in the background medium". Do you mean variations in density?**

Reply: The inhomogeneities in the temperature and wind field are responsible for the static/convective and dynamic instabilities, respectively, in the atmosphere which affect the wave propagation. We have included this statement in the revised manuscript (Page No. 3; Line No. 65-66).

**l. 75: "according to whether m^2 arises". I believe you meant "imaginary m^2"?**

Reply: We have modified the sentence in the revised manuscript (Page No. 4; Line No. 74).

**l. 124: What is the horizontal averaging of the SABER measurements?**

Reply: SABER views the Earth's limb in a direction of 90 degrees to the velocity vector of the spacecraft. The instantaneous field of view of the instrument is 2 km. The limb view offers high inherent vertical resolution due to the exponential decrease of pressure with altitude. Most of the radiance observed by SABER at a given tangent height originates within ±110 km of the tangent point. From this perspective, the 'horizontal' resolution of the measurements along the line-of-sight limb view is approximately 200 km.

**l. 142: What window sizes did you use?**

Reply: For our present case, the window size of SG-filter is taken as 400 pixels for OH, 350 pixels for 589.3 nm, 300 pixels for 866.0 nm, 300 pixels for 557.7 nm filter.
    On the other hand, the Gaussian filter has standard deviation ($\sigma$) of 20 pixels for OH, 20 pixels for 589.3 nm, 15 pixels for 866.0 nm, 15 pixels for 557.7 nm filter. Here, 1 pixel on the airglow images is nearly equal to 1.16 km. We have included this information in the revised manuscript (Page No. 6; Line No. 154-158).

**Equation 1: What are the values used for H and k_x?**

Reply: The values of $H = 5.4$ km and $k_x = 2\pi/\lambda_x = 0.084$ km$^{-1}$ ($\lambda_x = 75$ km). We have included this information in the revised manuscript (Page No. 6; Line No. 174-175).

**l. 185: What are the time offsets between acquisitions with different filters?**

Reply: The image time gap of different filters:

OH: 27 Seconds ⇒ 866.0 nm: 134 Seconds ⇒ 557.7 nm: 260 Seconds ⇒ 589.3 nm: 263 Seconds.

This is the cycle of the filter wheel operation. We have included this information in the revised manuscript (Page No. 4; Line No. 95-97).

**l. 185: "mean phase velocities". Are these observed phase velocities?**

Reply: Yes, these are observed phase velocities. We have mentioned that in the revised manuscript (Page No. 7; Line No. 212).

**l. 216: "there will not be any significant differences in the temperature". Well, I agree that significant differences between the SABER measurement and the temperature one hour later are not very likely, but the possibility can't be ruled out.**

Reply: Yes, it may be possible that the temperature changes after 1 hour. The change in the temperature within this short time scale is mainly due to wave perturbations, e.g., atmospheric tides. However, one hour is still a short enough time scale to assume that the change in the background temperature is very small and, thus, is not going to affect the analyses. We have included this information in the revised manuscript (Page No. 8; Line No. 242-245).

**l. 255: "In fact, clear signatures of wave activities were observed on other nights in OH airglow images over northern Germany." What about before and after the event you described in this paper? Was there any coherent structure observable in all four airglow bands?**

Reply: No, we didn't observe any other wave activities before and after the event. We have included this information in the revised manuscript (Page No. 8; Line No. 246).

**l. 264: "The SABER measurements indicated that the centroid height of OH airglow emission occurred near 87 km." Can you also provide estimates of the thickness of the OH layer (and potentially shape)? Because the vertical wavelength of the gravity wave is likely comparable to the thickness of the OH layer, even small changes in the thickness may have a large impact on the observability of the wave in OH airglow.**

Reply: The figure given below represents the OH volume emission rate (VER) profiles (both at 1.6 and 2.0 μm emissions) at SABER 2 measurement location.

[Figure]

From the above VER profiles, we can determine the thickness of the OH emission layer as 10.2 km. From Figure 7e, the average vertical wavelength of the wave structure is calculated to be 11.5 km ($m^2 = 0.3 \times 10^{-6}$ m$^2$). Thus, it is clear that the vertical wavelength of the wave structure is longer than the OH emission layer thickness. We have included this information in the revised manuscript (Page No. 11; Line No. 329-332).

In addition, we have analyzed SABER OH emission profiles for ±2 nights from nearby locations which is tabulated as follows:

| Date | Time (UTC) | Latitude (ºN) | Longitude (ºE) | Thickness of OH layer (km) |
|---|---|---|---|---|
| 2017-04-23 | 20:06:11 | 51.3812 | 6.12034 | 8.99 |
| 2017-04-24 | 20:19:49 | 50.5787 | 6.6311 | 10.05 |
| 2017-04-25 | 20:33:06 | 49.6448 | 5.17233 | 10.2 |
| 2017-04-26 | 20:47:06 | 47.385 | 6.60191 | 10.7 |
| 2017-04-27 | 21:00:33 | 45.5016 | 5.02261 | 10.2 |

It is to be noted that the OH emission layer thickness is similar. Hence, we believe that the small changes in the thickness will not have any significant impact on the observability of the wave in OH airglow. We have included this information in the revised manuscript (Page No. 10; Line No. 296-302).

**l. 273: "We believe that the perturbation wave amplitude will finally decide whether the structure observed in the OH airglow image is faint or not." Please explain. Are you arguing that the strength of the OH airglow emission ("faint or not") solely depends on the amplitude of the gravity wave?**

Reply: We have deleted the line in order to avoid confusion.

**l. 294: conductive**

Reply: We wrote "… conducive background condition…" in the manuscript (submitted version). We meant "favorable background condition".

**l 302: "It is well-known that the MIL tends to be quite stable within a few hours' time scale." Well, it depends on how you define "a few hours' time scale". For instance, tides can have a huge impact on the MLT *within a period of few hours*. Depending on the phase of the tide, mountain waves may propagate well into to the OH airglow layer or experience critical levels below.**

Reply: The MIL tends to be quite stable at least for 3-4 hours (e.g., Meriwether and Gerrard, 2004; Dao et al., 1995). We have modified the statement in the revised manuscript (Page No. 11; Line No. 342).

The tidally driven MIL tends to occur above 85 km in the MLT region. This MIL originates from large-amplitude tidal waves propagating into the mesosphere and their subsequent nonlinear interactions with gravity waves and tends to be quite stable.

**l. 309: delete "that" -> indicates the existence**

Reply: It has been deleted in the revised manuscript (Page No. 11, Line No. 349).

**l. 311: delete first "of" -> some energy**

Reply: We have deleted in the revised manuscript (Page No. 11, Line No. 351).

**l. 341: Please check the grammar of this and the following sentences.**

Reply: We have modified the sentence in the revised manuscript (Page No. 12, Line No. 379-381).

---

## Author Response (AR2)

**Reply to the Reviewer 1**

**Manuscript #: angeo-2021-48**

**Title: A case study of a ducted gravity wave event over northern Germany using simultaneous airglow imaging and wind-field observations**

**Thank you for your explanations and clarifications. I am sorry that I could not reply earlier.**

**I am still having difficulties following your arguments. In your comments you write:**

**"We suggest that the ducting layer (85-91 km) observed in the present case (Figure 7c & e) was weak and so inhibited free propagation within that particular altitude region. Only part of the energy of the wave could penetrate through the bottom of the duct layer and the other part of the energy will be reflected back downward. Being localized, the part of the wave continued to propagate upwards into the O2 and O(1S) emission layer at higher altitudes (represented schematically in Figure 8). Since only a part of the wave energy could enter in the duct region (85-91 km), the wave structure was observed to be faint in the OH airglow images."**

**Please forgive me my ignorance. According to your above explanation, upon reaching the bottom of the wave duct, the wave packet splits into three parts: one part being reflected and propagating downward (this part does not show up in Fig. 8), another part being trapped inside the duct, and the remaining part continuing propagating upward. Inside the duct, the waves are guided towards north-east and eventually cross your observation volume. But how do upward propagating waves which are detected in the O(1S) and O2 bands get there? If they crossed the thermal duct in the south-west where the ducting was supposedly weaker, the waves would have to propagate at a very shallow angle (nearly horizontally) in order to reach the observation volume at 94 km altitude (O2 emission). Given the large vertical wavelength, this doesn't seem to be likely. The other possibility I can think of – waves crossing the thermal duct within the observation volume – is not likely either, because these waves should show up in the OH band with approximately the same intensity as in the O(1S) band. Is there any obvious piece of information that I am missing?**

**Based on the data at hand there may not be a fully convincing explanation. But in the interest of scholarship you may want to acknowledge any inconsistencies in the explanation presented in the manuscript. In particular, the depiction of waves in Fig. 8 is misleading. I don't think waves can be refracted around the duct layer in this way.**

Reply: We thank the reviewer for the appreciation and comments. We have removed Figure 8 in order to avoid any confusion. Based on the reviewer's suggestion, we have also included a paragraph discussing the limitation of this paper (Page 12, line 368-377). We hope that the reviewer will appreciate our effort.